# Karst Landscape Governance in the Guilin World Heritage Site, China

Guizhen He [1,2,*], Mingzhao Yu [1], Xiang Zhao [1,2], Lei Zhang [3] and Lina Shen [4]

1 State Key Laboratory of Urban and Regional Ecology, Research Centre for Eco-Environmental Sciences, Chinese Academy of Sciences, Beijing 100085, China; mzyu@rcees.ac.cn (M.Y.); zhaoxiang194@mails.ucas.ac.cn (X.Z.)
2 College of Resources and Environment, University of Chinese Academy of Sciences, Beijing 100149, China
3 School of Environment and Natural Resources, Renmin University of China, Beijing 100872, China; leizhang66@ruc.edu.cn
4 Key Laboratory of Karst Dynamics, MNR & Guangxi, Institute of Karst Geology, Chinese Academy of Geological Sciences, Guilin 541004, China; shl8242@163.com
* Correspondence: gzhe@rcees.ac.cn

**Abstract:** Sustaining karst landscape areas in World Heritage Sites under increasing human pressures and climate change is an emerging challenge. Growing evidence has highlighted the transition from traditional government-oriented regulation to the collaborative governance of different stakeholders in governing karst landscape resources. However, the complexity and dynamics of karst landscape policy and stakeholder networks are poorly understood. This study combined a legislative analysis, network analysis, and public survey to explore effective methods of karst landscape conservation in the Guilin World Heritage Site, China. The policy analysis showed that various national laws and local regulations have been enacted in China, but these regulations often focused on individual aspects of karst–human interactions. The network analysis indicated the complexity and relationship of networks in karst World Heritage Site governance at the national, provincial, and municipal scales. The majority of questionnaire respondents (65–89%) reported a medium and high level of karst landscape governance effectiveness. The insights in the present study may be valuable for other karst World Heritage Sites facing complex challenges, especially in developing countries.

**Keywords:** karst landscape; social network analysis; policy analysis; stakeholder; nexus governance; Guilin World Heritage Site





## 1. Introduction

Karst landscapes, which have high cultural, historical, scientific, aesthetic, and recreational value for humankind, are widespread in Asia, Europe, Central and North America, and the Caribbean [1–3]. Karst landscape conservation is intrinsically interlinked with at least eight Sustainable Development Goals (SDGs) of the United Nations 2030 Agenda for Sustainable Development. Until January 2022, there were 75 World Heritage Sites (WHS) with carbonated karst and 1 WHS with evaporite karst; the two categories (carbonated and evaporitic) totaling 841,422 km$^2$ [4]. Karst has been affected by climate change and human factors such as population growth, urbanization, agricultural expansion, water extraction, and mining [5–8]. There are complex stakeholders within the geographical unit of karst heritage. The state, provinces, municipalities, and other stakeholders have been seeking to coordinate and cooperate to achieve inter-regional and basin-based karst heritage conservation [9–11]. Some previous publications promoted a holistic approach of karst management [12]. However, existing studies do not reflect the complexity and dynamics of karst heritage governance and stakeholder networks. Exploring effective methods for karst landscape preservation from the perspective of policy and governance is essential to achieve sustainable karst heritage conservation [13].

Karst heritage policies and management have received only indirect attention in the past. Early studies often focused on the distinctive geomorphology, geochemistry, and hydrology of karsts, rather than their management [14,15]. The importance of karst heritage conservation was highlighted by the International Union for the Conservation of Nature and Natural Resources (IUCN) in 1997 and updated in 2022 [16]. In recent decades, several countries have launched technical projects to mitigate and restore karst landscapes, and several ecological management and control examples of rocky desertification in karst areas were proposed [17]. During this process, government played an important role. Engineering measures adopted in karst areas, for example, include water body restoration in North America, the comprehensive control and management of rocky desertification in the USA, and tourism in the karst mountains in Switzerland [18].

In China, researchers began to analyze karst degradation in the 1980s. An increasing number of scientists made efforts to explore measures for karst ecosystem restoration and find various management models for karst conservation [8,19,20]. Typical patterns have been presented, such as a forest restoration model; the development of herbivorous animal husbandry, namely, the "grass + livestock + biogas" model in karst areas; a soil and water conservation model; an eco-agriculture model; the establishment of ecological reserves and the development of tourism; and a comprehensive management model [21–23]. However, few studies have been published in the fields of karst conservation policy, management, and governance [9].

Karst conservation requires the coordination and involvement of multiple stakeholder groups. As there are diverse wants and needs among different stakeholders, research on the intricacies and dynamics of karst management networks needs to be expanded using a nexus approach or an integrated perspective. Local residents are both direct prime makers and beneficiaries in karst landscape changes. Increasing public awareness of the need to protect our endangered karst landscape resources, especially in developed countries, has promoted the development of karst-specific environmental governance. Recently, several approaches have been proposed to assess the degree of human-induced disturbances in karst areas [24–26]. These studies are essential for the proper management of karst landscape resources. While these efforts are certainly not futile, their effectiveness will likely only become visible through transformative actions, including restoring the productivity of degraded land, coordinating and planning across sectors, policy coherence and harmonization of national strategies and plans, the use of participatory frameworks, and the establishment of partnerships [13,17,27]. To truly understand the relationships in karst management, it is also important to consider the other side of the relationship: the stakeholder's perspective.

In China, governments have increased their involvement in the planning and management of karst landscapes due in part to resource utilization. However, the transition to a participatory approach is desirable in the context of environmental governance. In practice, at least three levels of government, national, provincial, and municipal, as well as other stakeholders, have attempted to coordinate and work together to support interregional and watershed karst landscape conservation. This study aims to understand how karst heritage conservation is being implemented in China. Selecting Guilin City as the study area, we described Chinese legislation and policies on karst-landscape-related conservation and the key governmental institutions and other organizations involved in managing karst landscapes. Social network analysis (SNA) [28] was used to demonstrate relationships within the multi-sectoral network for karst landscape conservation in Guilin. The effectiveness of karst landscape conservation was identified and assessed using a local public perception survey. This study provides the reader with a broad understanding of the regulatory policies and complex governance linked to karst landscape conservation in China.

## 2. Methodology

### 2.1. Study Area

Guilin City is located in the northeastern part of Guangxi, China (Figure 1). Nearly 2452 km$^2$ of karst landscape is distributed within 20–60 km along the Lijiang River in Guilin, with well-known fenglin or peak forests (isolated towers) and fengcong or peak cluster forests (linked-base towers) [29,30]. In terms of the spectacular landform types and landscape, as outstanding examples of landscape evolution and karst development of significant global value, the Phase II South China Karst, was inscribed on the World Heritage List at the 38th session of the World Heritage Committee (WHC) on 23 June 2014. The four specifically selected areas were Guilin Karst and Huanjiang Karst in Guangxi, Jinfo Mount Karst in Chongqing, and Shibing Karst in Guizhou [31].

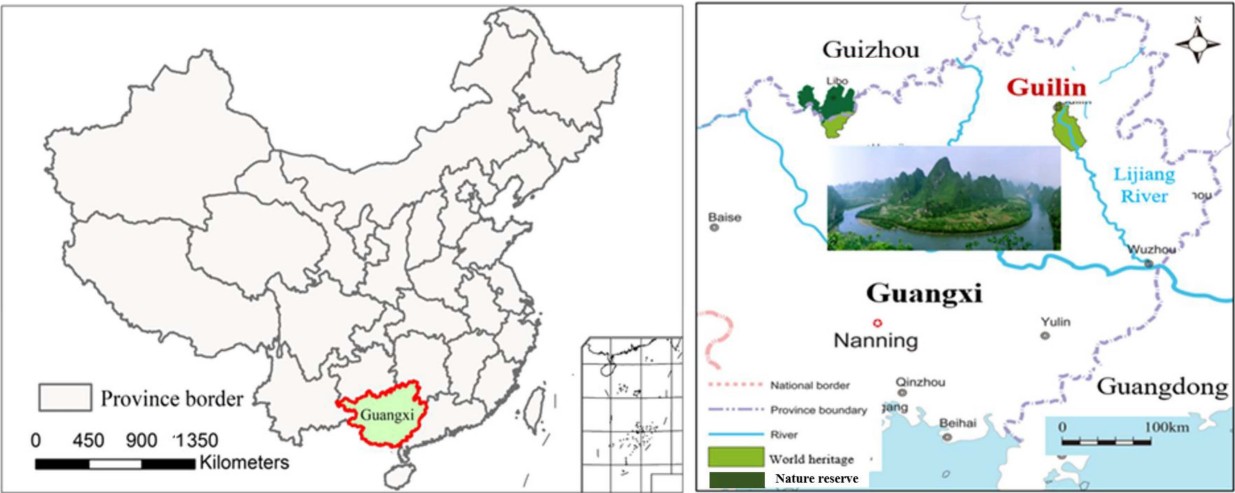

**Figure 1.** Map of the study area in Guilin, Guangxi, China.

As a world-famous tourist destination, Guilin was visited by over 138 million international and domestic tourists and contributed RMB 18 billion in tourism revenues in 2019, which decreased due to COVID-19 in 2020 and 2021. Several ecological projects including the Grain to Green Project, the Natural Forest Protection Project, and the Karst Rocky Desertification Restoration Project have been implemented in the Lijiang River Basin area of Guilin since 1999 [32]. Moreover, Guilin City was selected as one of three pilot cities for constructing an innovative demonstration zone as part of the 2030 Agenda for Sustainable Development with the theme of "sustainable use of landscape resources", which was a national program promoted by the Ministry of Science and Technology and approved by the State Council in 2018.

### 2.2. Data and Analysis Methods

In this study, we used a combination of: (1) legislative analysis, (2) Gephi software of social network analysis [28], and (3) a questionnaire survey to collect information. Firstly, we collected secondary data through document review and survey methods. The national laws and regulations related to karst conservation and management were collected from 39 official websites including the National People's Congress, the State Council, and ministries and departments. The provincial regulations were collected from 32 provincial governments, departments, and bureaus. The city-level regulations and policies were collected from 30 municipal-level governments and bureaus. Secondly, the institutions involved in karst conservation were identified based on the functions, responsibilities, and practices prescribed in the laws and regulations. The official websites of different national ministries, provincial institutions in Guangxi, local institutions in Guilin, and other institutes, universities, and non-governmental organizations were visited. The duties,

documents, projects, and activities related to karst conservation were recorded and compiled. Stakeholder mapping was performed based on the above information. The Gephi social network analysis software was used to analyze the governance network for karst conversation and management [28].

Finally, a survey was conducted in Guilin using a specifically designed questionnaire. Residents from Lingchuan County, Yangshuo County, Xing'an County, and the city center of Guilin took part in the online survey to gather opinions related to landscape governance. The questionnaire was divided into two sections. The first section queried the respondents' demographic characteristics (gender, age, income, etc.), while the second section assessed karst landscape governance in Guilin. The questionnaire was distributed to 1200 local residents via a single online survey system in December 2021. Respondents in Guilin were selected using a random sampling method and were 18 years of age or older. Only completed questionnaires were collected, which resulted in 1040 valid responses (a response rate of 86.7%). The respondent IDs, personal information, and question scores were recorded in an Excel file. The data were analyzed with the statistical software SPSS 22.0. Differences among samples in the percentage, mean, and standard deviation (SD) were analyzed via descriptive analysis. A Spearman's correlation analysis was conducted to determine public perceptions and their sociographic variables. A Mann–Whitney U test and Kruskal–Wallis test were applied to determine the statistically significant differences between groups. We set the level of statistical significance at $p < 0.05$ (95% confidence interval).

## 3. Multilevel Policy Related to Karst Heritage Conservation in China

### 3.1. Laws and Administrative Regulations

The capacity of karst heritage conservation depends on the governance system shaping the laws, ordinances, standards, policies, formal and informal rules, norms, and decision-making in individual and collective actions. The Convention Concerning the Protection of the World Cultural and Natural Heritage, 1972, is one of the most important global conservation instruments for identifying and protecting natural and cultural heritage sites worldwide. China joined the Convention in 1985. Faced with increasing pressure from environmental nonprofit groups and the public, new laws and regulations in China have been developed to protect environment and natural resources since the late 1970s. After the trial Environmental Protection Law in 1979, the national government passed laws protecting cultural relics in 1982 and revised these laws in 2002. During the 1980s and 1990s, national environmental laws and regulations were established to protect natural resources and the environment, such as water and air (see Table S1). However, China currently has no unified or specialized karst regulations or management law.

Karst heritage conservation management is divided among relevant laws, regulations, and policies at the national and local levels. By the end of 2022, 40 national laws, 47 administrative regulations of the State Council, 47 local regulations, 36 government rules of Guangxi, nine local regulations, and 35 government rules of Guilin had been promulgated and implemented (Figure 2). These laws and administrative regulations/rules related to karst conservation and management involved nine categories such as ecology and environmental protection, natural resources, and culture and tourism. Among them, there were two laws closely related to karst conservation: The Law on Protection of Cultural Relics, 2002, and the Law on Intangible Cultural Heritage, 2011. One similar administrative regulation of the State Council was Regulations on the Protection of Paleontological Fossils, 2010 (revised in 2019). The Departmental Regulation on the Protection of Geological Remains 1995 was established by the former Ministry of Geology and Mineral Resources (now the Ministry of Natural Resources). Most laws correspond to natural resources (17), followed by environmental protection (9). Natural resource laws and regulations cover the land, geological and mineral resources, wildlife resources, forestry, grassland, wetland, water resources, fishery resources, etc. The latest Wetlands Conservation Law enacted on 1 June 2022 is closely related to karst landscape conservation. The environmental protection domain involves water, air, waste, soil, toxic substances, and chemicals. Other indirectly

related laws include the Constitution; Law on Legislation, 2000 (revised in 2015 and 2023); and Criminal Law, 1979 (revised 11 times until 2022). In recent years, multi-focus laws, regulations, and ordinances that relate to the impacts of new developments on karst landscape, groundwater, and the integrity of new urban plans are becoming increasingly common.

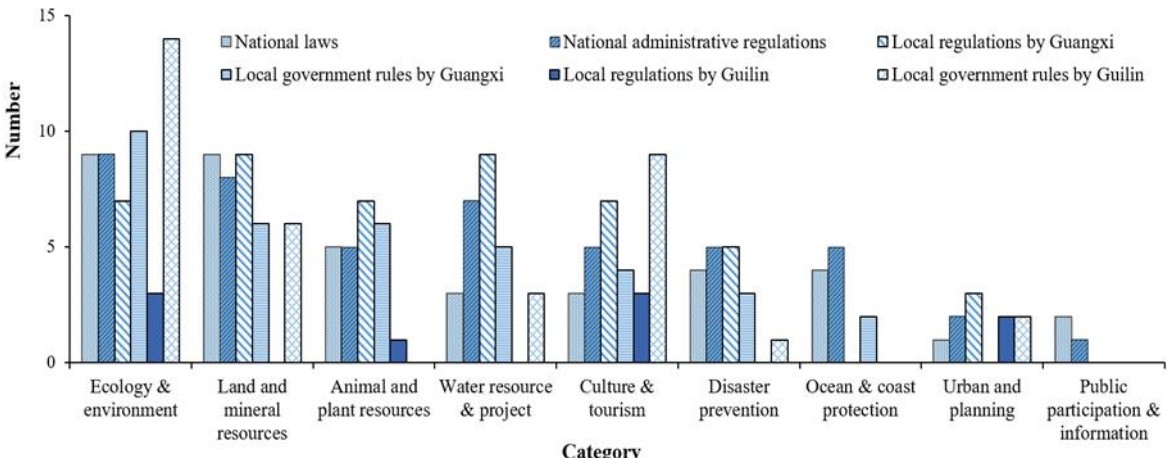

**Figure 2.** Categories of laws and administrative regulations related to karst conservation and management at the national and local levels until March 2022.

Karst heritage regulations are not universal because governments are often not given a sufficiently wide range of tools with which to manage karst. The available tools are typically limited to regulations, ordinances, and plans for land, minerals, scenic spots, etc. At the provincial level, most local regulations correspond to land/mineral resources and water resources, and most local governmental rules relate to ecology and environment. Relevant regulations include the 2011 Regulation of Guangxi Zhuang Autonomous Region on Lijiang River Basin Ecological Protection, 2020 Regulation of Guangxi on Stalactite Resources Protection (first promulgated in 2002 and revised in 2004, 2009, 2016, and 2020), 2016 Regulation of Guangxi on Scenic Spots Management (first promulgated in 1999 and revised in 2004, 2010, and 2016), and 2014 Administrative Measures of Guangxi for the Protection of Lingqu. At the city level, the number of local regulations is the lowest with nine in total. Fourteen local governmental rules relate to ecology and the environment, followed by nine culture and tourism rules (Figure 2). The Regulation of Guilin on Stone Carving Protection and the Regulation of Guilin on Lijiang River Scenic Spots Management were implemented in 2017 and 2020, respectively. On 1 January 2022, the Regulation of Guilin on the Sustainable Use of Karst Landscape Resources and Regulation of Guilin on Lingqu Protection were enacted, representing a new milestone for karst landscape conservation in Guilin City.

*3.2. Sectoral and Local Regulations and Administrative Documents*

In China, karst heritage issues are often addressed by ministerial, provincial, and municipal administrative regulations. Differences in social and economic status, as well as physical landscapes, between different provinces and cities usually influence the selection of regulatory techniques adopted in each province and city. However, several regulatory techniques are used more frequently than others, including local regulations, planning and zoning guidelines, and management rules. In terms of karst conservation, the government departments at different levels have issued many sectoral ordinances and normative documents. At least 16 national ministries, 16 departments of Guangxi, and 18 bureaus of Guilin enacted 213, 238, and 223 departmental ordinances and normative documents, respectively, targeting the ecological and environmental protection of karst terrains (Figure 3).

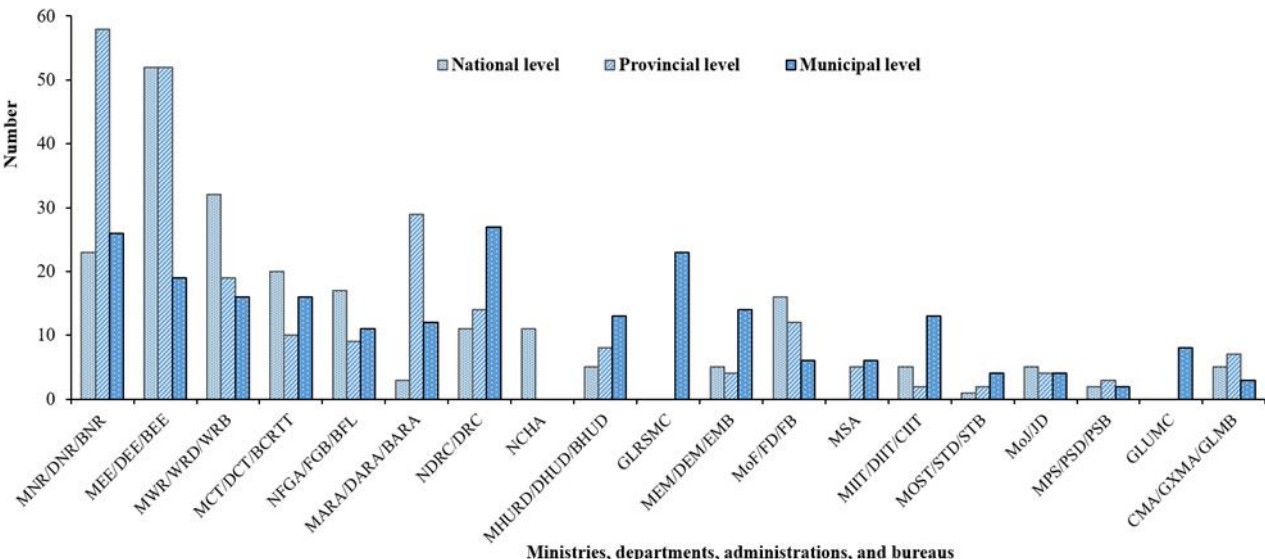

**Figure 3.** Ordinances and normative documents enacted by Chinese ministries and administrations until March 2022 (see abbreviations of the full list in the Supplementary Materials).

At the national level, the Ministry of Ecology and Environment (MEE), Ministry of Water Resources (MWR), and Ministry of Natural Resources (MNR) are the main departments for promulgating departmental regulations related to karst conservation and management. The Ministry of Culture and Tourism (MCT) issued six ordinances and 14 normative documents that are closely related to karst landscape management. Administrative Measures for the Protection of World Culture Heritages is the most closely related. At the provincial level, the Department of Natural Resources (DNR), Department of Ecology and Environment (DEE), and Department of Agriculture and Rural Affairs (DARA) in Guangxi are the first three departments enacting departmental polices related to karst conservation. At the Guilin City level, the Bureau of Natural Resources (BNR), Guilin Lijiang River Scenic Area Management Committee (GLRSMC), and Bureau of Ecology and Environment (BEE) in Guilin issued 26, 23, and 19 departmental polices and documents related to karst conservation, respectively. As a rule, these laws and regulations impose mandatory obligations controlling the negative impacts from human activities.

In essence, karst landscape problems from karst–human interactions usually require solutions from more than one field and sector. However, local regulatory organizations often have more narrowly focused power and responsibility, which creates difficulties in taking an interdisciplinary approach. In this situation, bodies without any actual regulatory power, such as karst-related research institutes and geological surveys, can act as supporters and catalysts for policy making and information/data providers for shaping these solutions. While karst landscape conservation is inherently a complex issue, we can still benefit from a regulatory approach, even though some flaws in existing karst protection regulations may hamper effectiveness. In some cases, this failure is a result of vaguely defined management goals, a lack of appropriate tools, poor execution, and weak or nonexistent enforcement.

## 4. Network and Perceived Effectiveness of Karst Heritage Governance

Karst heritage governance approaches aim to achieve meaningful collaboration and participation by different actors in conservation policy, science, and practice. It is important to find possible approaches for adaptive decision-making and management with multi-level perspectives to promote in situ resource ownership and use [33]. The governance of karst heritage concerns multi-sectoral stakeholder networks, with many public, for-profit, and non-profit organizations involved, notably national and local governments, scientific institutions and universities, enterprises, the media, the community, the village committee, and the general public.

### 4.1. Stakeholders Involved in Karst Heritage Governance

To explore how actors in a multi-sectoral network coordinate with each other, we empirically identified stakeholders' network capital and centrality by analyzing the relationships within karst heritage conservation in Guilin. The study was built using official websites belonging to national, provincial, and municipal institutions and their online documents (e.g., news, policies, plans, and annual reports) alongside social network analysis. The stakeholders identified in the Guilin karst heritage conservation covered governments, research institutes, and universities, as well as non-governmental organizations (NGOs). The results indicated that actors used six types of connections in their coordination efforts: legal, regulatory, technical, communication, external links, and internal links. For example, the National People's Congress had legal links with other actors, but the State Council had a regulatory relationship with other actors. Government efforts to protect karst landscapes have proven to be significant catalysts of social, economic, and environmental development [20]. The Gephi software can identify who shares common interests in networks and complex systems, and was used in this work to conduct node and edge analyses, as well as study the interactions between different actors based on various metrics (Table 1). There were 176 nodes and 1321 edges in total. The whole network analysis clarified at least one actor in the core from each governmental level, demonstrating the important coordination structure in karst landscape governance for governments and other actors.

**Table 1.** Multi-level actors network of karst heritage governance in Guilin (for abbreviations, please see Table S2 in the Supplementary Materials).

| Network | No. of Nodes and Edges | Metrics Statistics | Core Actors and Percentages |
|---|---|---|---|
| Full network | Nodes: 176 Edges: 1321 | Average Degree: 15.011 Graph Density: 0.086 Modularity: 0.268 Average Clustering Coefficient: 0.345 Average Path Length: 2.149 | Core: 20 actors (11.36%) Core national actors: MNR, MCT Core provincial actors: GXPG Core municipal actors: GLRSMC Core non-government actors: IPCK |
| National agencies | Nodes: 76 Edges: 452 | Average Degree:11.895 Graph Density: 0.159 Modularity: 0.210 Average Clustering Coefficient: 0.300 Average Path length: 2.001 | Core: 11 actors (14.47%) Core national actors: MCT Core non-government actors: IRCK |
| Institutions in Guangxi | Nodes: 51 Edges: 243 | Average Degree: 9.529 Graph Density: 0.191 Modularity: 0.146 Average Clustering Coefficient: 0.087 Average Path Length: 2.017 | Core: 9 actors (17.65%) Core provincial actors: GXDNR, GXDEE, GXDCT |
| Institutions in Guilin City | Nodes: 49 Edges: 331 | Average Degree: 12.291 Graph Density: 0.240 Modularity: 0.107 Average Clustering Coefficient: 0.129 Average Path length: 1.119 | Core: 9 actors (18.37%) Core municipal actors: GLRSMC, GLBCRTT, GLBNR |

A visual depiction of all actors and their coordination shows core actors and their connections with others (Figure 4). By analyzing the hub and authorities of the whole network, the results indicate a mix of actor levels. The Ministry of Natural Resources (MNR) and Guilin Lijiang River Scenic Area Management Committee (GLRSMC) are the most salient actors in the network, indicating their critical role in regulation and coordination, especially in facilitating regulation and communication to support management measures and projects.

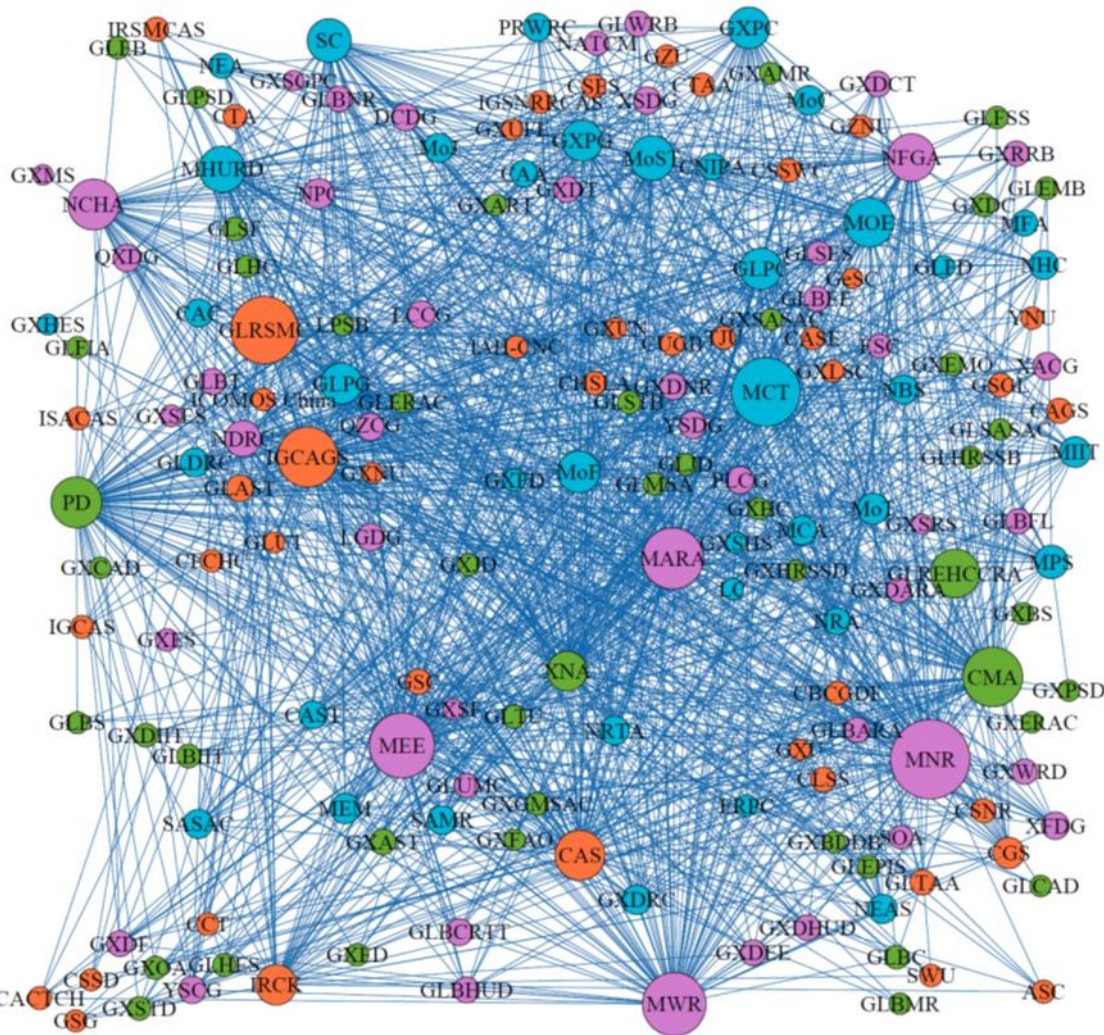

**Figure 4.** Network analysis of actors in Guilin's Karst Heritage Governance. The size of the dots represents the role in karst conservation, while the color is automatically generated and does not hold any specific meaning.

The Ministry of Natural Resources (MNR) was established in 2018 according to the Institutional Reform Plan of the State Council approved by the First Session of the 13th National People's Congress. The MNR was formerly known as the Ministry of Land and Resources; the National Forestry and Grassland Administration, National Park Administration, and National Oceanic Administration are affiliated institutions. The main responsibilities of the MNR are to supervise the development, utilization, and protection of natural resources, establish a spatial planning system and supervise its implementation, and to be responsible for the management of surveying, mapping, and geological exploration industries. Karst landscape protection is also one of the MNR's responsibilities. At the national level, the Ministry of Culture and Tourism (MCT) is also an institution directly responsible for karst heritage management. The MCT was restructured in 2018 based on the former Ministry of Culture. The duties covered by this institution include cultural heritage and intangible cultural heritage management, cultural relics and archaeology, tourism planning, and policy development. At the city level, the Guilin Lijiang River Scenic Area Management Committee is the legal management agency assigned by the Guilin Municipal People's Government and was established on 1 August 2017. This agency is directly responsible for managing the Lijiang River Scenic Area, which is mainly distributed in the Yanshan District of Guilin, Lingchuan County, and Yangshuo County, with a total planning area of 1159.4 km. With the overall goal of "letting the beautiful Lijiang River

be preserved forever", the GLRSMC incorporated concepts of whole planning, precise treatment, systematic protection, and scientific utilization to protect water, mountains, and the karst landscape along the Lijiang River.

At the national level, there are 76 nodes and 452 edges (Table 1). The national cores are the national government ministries due to their stated regulatory and coordinating functions. The MCT and MNR are key authorities through which karst regulations are proposed/finalized/approved after they receive information (Figure 5a). This structure is in keeping with a horizontal management approach and the national government's whole-of-government approach for karst landscape heritage conservation coordination. The Institute of Karst Geology, Chinese Academy of Geological Sciences (IGCAGS), founded in 1976, is a governmental-oriented scientific research institution directly under the purview of the Chinese Academy of Geological Sciences of the MNR. This institute focuses on basic and applied karst geological research in China, especially in southwestern China, and is mainly engaged in six areas of study: karst dynamics and global change, karst ecosystem and rocky desertification control, karst water resource surveying, evaluation, development, and utilization, karst collapse geological disaster prevention and control, karst landscape and cave tourism resource evaluation, and carbonate oil and gas reservoir research. In recent years, the IGCAGS conducted monitoring, surveys, and studies related to karst landscape projects and management in Guilin. The International Research Center on Karst under the Auspices of UNESCO (IRCK) is a core non-government actor. The IRCK was established jointly by UNESCO and the former Ministry of Land Resources in 2008. The responsibilities of this entity include carrying out technical, scientific, and theoretical research on modern karst science, coordinating and organizing international and regional cooperation projects, the international exchange of scientific and technological information and policy information, and international karst training. Since its establishment, the Center has held 11 international conferences and conducted academic exchanges with more than 200 experts from more than 40 countries.

At the provincial level, the results showed a uniformity of actors. The governmental departments were the key hub, and others were related authorities (Figure 5b). At the municipal level, the results also indicated a uniformity of actors, while the hub and authorities were GLRSMC, Bureau of Culture, Radio, Television and Tourism (GLBCRTT), and Guilin Bureau of Natural Resources (GLBNR) (Figure 5c). As a management agency directly led by the Guilin Municipal People's Government (described previously), GLRSMC leadership consists of 13 members, among whom six are full-time leaders. The other seven members are adjunct leaders working for Guilin Municipal People's Congress, Guilin Municipal People's Government, Guilin Public Security Bureau, and the district/county governments of Lingchuan, Yangshuo, and Yanshan. The GLRSMC is responsible for scientifically and holistically protecting, utilizing, managing, and operating the resources of the Lijiang River Scenic Area by coordinating the interests of all parties. The latest Overall Plan of Guilin Lijiang River Scenic Area (2022–2035) was developed and implemented to rationally utilize the resources of the Lijiang River Basin's green development. The GLBCRTT is an institution responsible for cultural and tourism policy, plans, and industry issues in Guilin. This institute conducts investigations of karst cultural relics and archaeological resources, formulates local plans and policies, constructs key cultural and tourism facilities and major projects, and organizes international events in Guilin. The GLBNR is a municipal agency responsible for natural resource planning (such as land, mineral, forest, water, and wetland), investigation, exploration, utilization, supervision, and ecological remediation. The results indicate the inter-governmental coordination and implementation foci of the local governments and also agree with the types of connections that each government develops for karst landscape governance. Our findings present the significance of governments in coordinating multi-sectoral karst management networks. Therefore, partners could consider how to re/structure an administrative organization network to fulfill their different strategic goals.

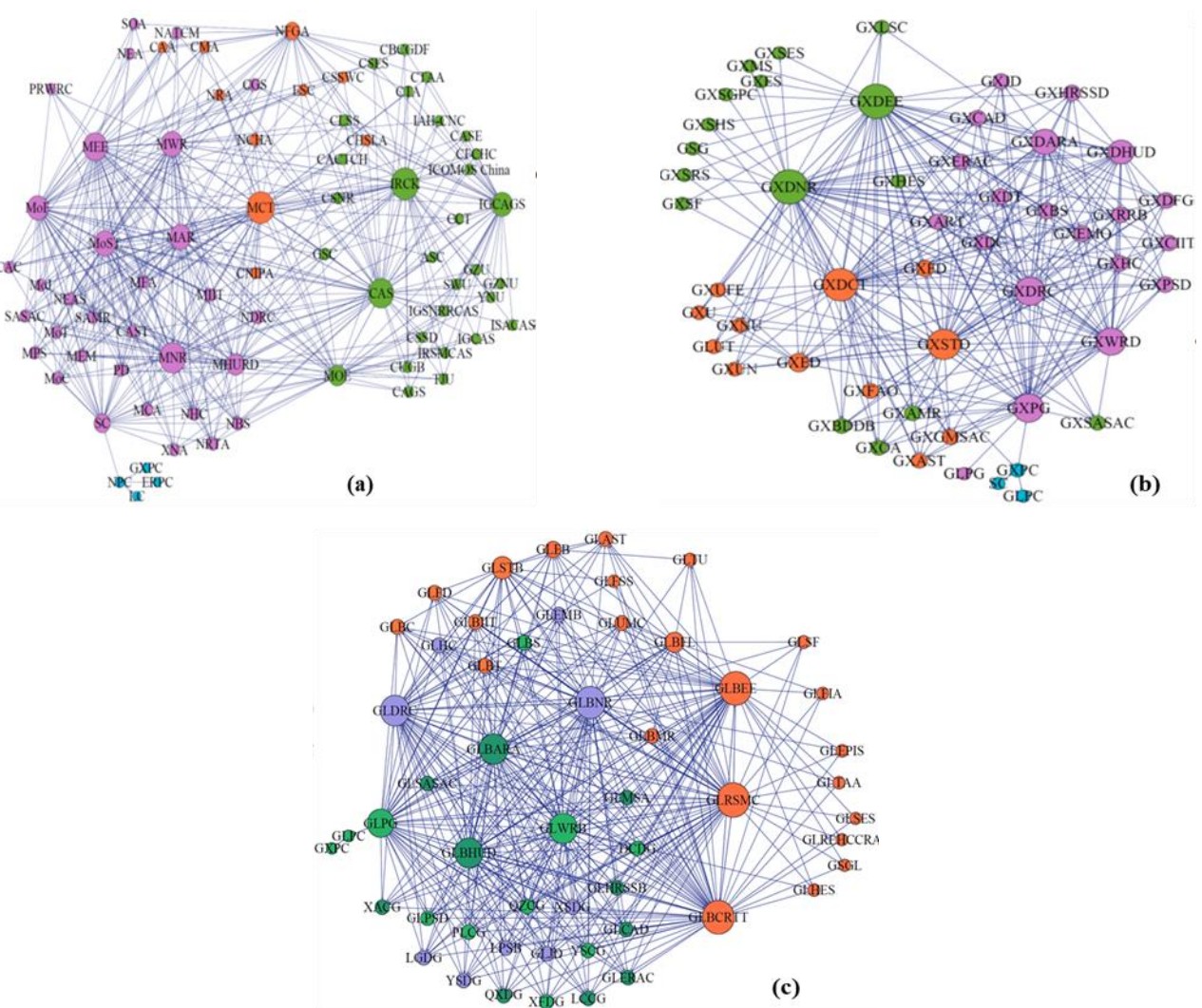

**Figure 5.** Network analysis among actors for karst landscape governance at different levels. (**a**) National level in China, (**b**) provincial level in Guangxi, (**c**) municipal level in Guilin City. The size of the dots indicates the actors' roles in karst conservation, while the color is automatically generated and does not hold any specific meaning.

*4.2. Mapping Stakeholders' Roles in Guilin Karst Governance*

In this study, the "stakeholder" concept includes both local residents and people with an economic interest in the location (the industry and resource users), as well as governments and agencies at all levels. The influence varies widely between each group of stakeholders. A stakeholder mapping grid was created through a standard 2 × 2 matrix [34], representing influence and interest dimensions in terms of a desired outcome, defined as a person "who has interest in and influence over karst landscape governance resources for supporting sustainable development in the Lijiang River Basin, Guilin" [35]. Stakeholder influence and interest used a ranking scale from one (no influence/interest) to five (extremely influential/interested). The four groups of stakeholders had different functions. All stakeholders involved were classified as rule setters and regulators, key implementers, data suppliers and crowds, and subjects in karst landscape governance (Figure 6).

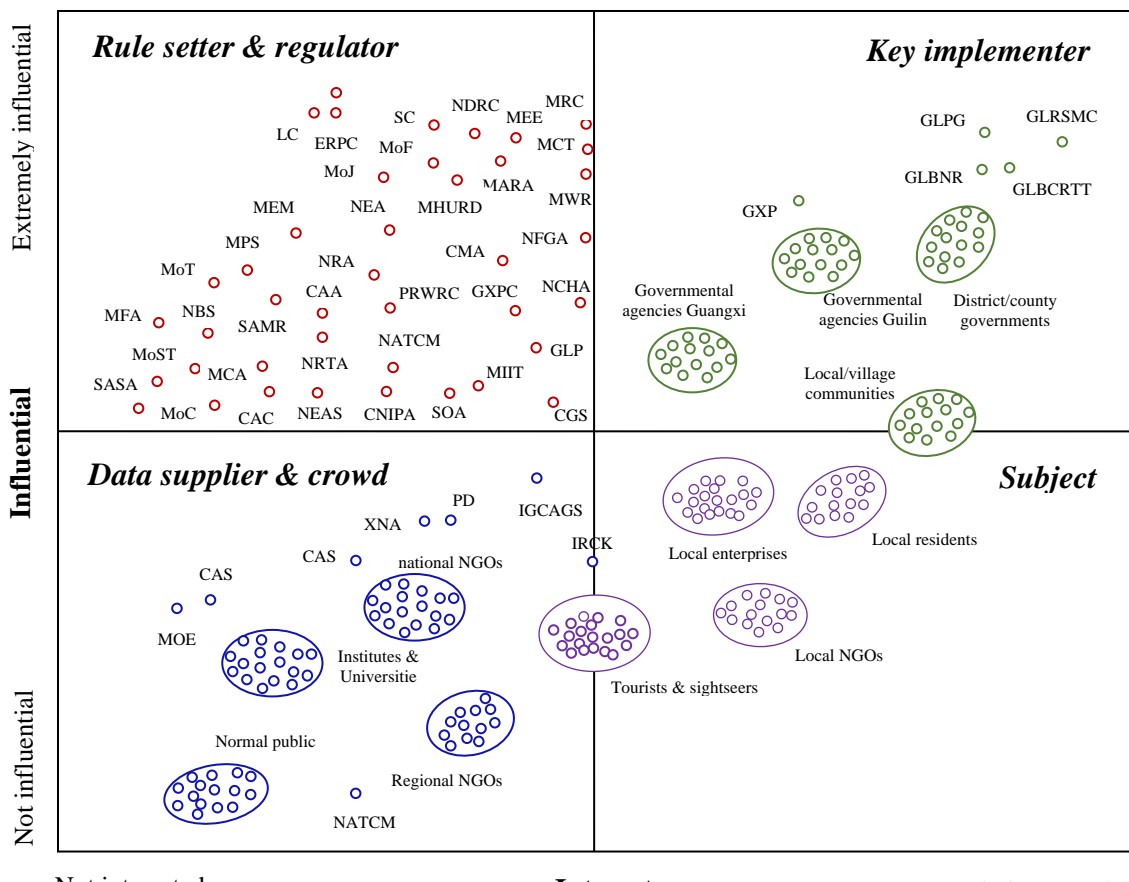

**Figure 6.** Mapping stakeholder interest in and influence on karst management in Guilin. The full list of acronyms is given in Table S2 of the supplementary materials.

Rule setters and regulators with high influence on the governance of karst landscapes but low interest in implementing measures in the Lijiang River Basin area were mostly represented by governmental organizations in the natural resources, culture and tourism, and environmental sectors (MNR, MCT, and MEE). Normally, these actors play important roles in promulgating national laws, departmental ordinances, plans, and policies to provide guides for local karst landscape management. Key implementers or actors with influence on and interest in governing karst landscape resources are the most influential stakeholders in achieving the proposed goals and acting with high interest. These stakeholders include not only the national ministries related to natural resources, the environment, forest, and water, but also local governments and official agencies with direct responsibility for the regulation and management of natural resources, the environment, water, land, and tourism such as the Guilin People's Government, Guilin Lijiang River Scenic Management Committee, and Bureau of Natural Resources, among others. Interestingly, the natural resource and culture sectors, represented by the Ministry of Natural Resources, appeared to be both regulators and key implementers, possibly due to their role as both managers and project designers in karst landscape conservation and governance.

The subjects were the stakeholders with high interest in the proposed outcome but low influence in achieving it. Here, the subjects were represented by local enterprises, local/village communities, and NGOs. Local enterprises and communities normally cause degradation of, and pose a risk to, the karst landscape. Village communities use land and water resources in karst regions for their livelihoods. Because of their lack of influence over management decisions and regulations, these communities are subject to the decisions made by rule setters and key implementers. These communities can increase their influence by

uniting themselves with more influential key players to help achieve the desired outcomes. The data suppliers and crowds with low interest and low influence in achieving the desired outcome might be ignored, as these actors tend to be bystanders in the process. Data suppliers are normally educational and scientific institutions, institutes, universities, and national NGOs. The crowd mostly consists of stakeholders who are not responsible for the direct use or management of any particular karst landscape resource. Here, the mapping presented some actors who were close to the boundary areas of the mapping grid, e.g., tourists (Figure 6). Thus, the location of the stakeholders needs to be interpreted with care. Notably, it is also important to monitor the crowd and, indeed, all stakeholder groups for shifts in interest and influence with changing circumstances over time.

### 4.3. Assessing Governance and Stakeholders' Roles in Karst Heritage Conservation

Assessments of karst conservation outcomes are intricately linked with public perceptions of management and governance, which produce evidence for how well karst governance regimes operate and where problems exist. The most important aspects in previous studies were considered in the present analysis [33,36,37]. In our survey, we invited the respondents to present their perceptions of karst heritage governance by focusing on the following: (a) social and political awareness of and environmental concern for karst conservation issues, (b) the roles of governmental laws and policies at multiple scales, (c) the perceived effectiveness of governmental policies/projects, (d) participation and coordination related to the use, management, and protection of karst resources among different stakeholders, and (e) trust in institutions that regulate local communities and other stakeholders. The scales ranged from one to five and were presented from the lowest to highest level. Overall, participants' perceptions on the five aspects of karst landscape governance were positive. The mean scores of these aspects were between 3.52 and 4.18, while the standard deviation (SD) ranged from 1.21 to 1.53 (Table 2). The highest score was recorded for the roles of governmental law and policy. Participation and coordination in karst landscape resources among different stakeholders obtained the lowest mean score of 3.52. Approximately 23.65–6.78% of the respondents rated the five aspects with the highest ranking, while more respondents (41.45–52.31%) selected a high level for different variables of karst heritage governance in Guilin. Only a minority of participants responded with the low and lowest levels of karst landscape governance (2.11–6.27%).

**Table 2.** Participants perception of karst heritage governance based on five aspects ($n = 1040$).

| Ranking | Awareness and Concern | Roles of Law and Policy | Perceived Effectiveness | Participation and Coordination | Trust in Institutions |
|---|---|---|---|---|---|
| The lowest | 0.38% | 0.38% | 0.77% | 1.59% | 0.50% |
| Low | 3.46% | 1.73% | 4.23% | 4.68% | 3.38% |
| Medium | 20.19% | 14.42% | 20.00% | 20.28% | 17.89% |
| High | 52.31% | 46.73% | 50.58% | 41.54% | 41.45% |
| The highest | 23.65% | 36.73% | 24.42% | 31.91% | 36.78% |
| Mean scores | 3.95 | 4.18 | 3.94 | 3.52 | 4.11 |
| SD | 1.21 | 1.49 | 1.53 | 1.36 | 1.39 |

A Spearman correlation analysis showed that the respondents' demographic characteristics such as living place and sex had no significant correlation with the perceived five aspects (Table 3). There were significantly positive relationships between age and the participants' perceptions of roles of law/policy ($p < 0.001$), effectiveness ($p < 0.01$), and trust ($p < 0.001$). There was a significantly positive correlation between the respondents' education levels and perceived effectiveness ($p < 0.05$) and a significantly negative correlation between the respondents' education level and trust in institutions ($p < 0.01$). There were significantly positive relationships between the participants' monthly incomes and perceptions of awareness/concern ($p < 0.01$) and trust ($p < 0.05$). There were also significantly positive relationships between the participants' professions and perceptions

of awareness/concern ($p < 0.01$) and participation ($p < 0.05$). There were, moreover, significantly positive relationships between the distance to Guilin Karst World Heritage Site and perceptions of awareness/concern ($p < 0.01$) and roles of law/policy ($p < 0.01$) and a significantly negative correlation between the distance to Guilin Karst World Heritage Site and trust in institutions ($p < 0.01$). Lastly, there were significantly positive relationships between the living time of respondents and their perceived effectiveness ($p < 0.001$) and participation ($p < 0.001$).

**Table 3.** Spearman correlation analysis between public perceptions and sociographic variables.

| Variables | Test Results | Awareness and Concern | Roles of Law and Policy | Perceived Effectiveness | Participation and Coordination | Trust in Institutions |
|---|---|---|---|---|---|---|
| Age | Correlation coefficient | 0.016 | 0.119 *** | 0.101 ** | 0.053 | 0.120 *** |
| | Sig. (2-tailed) | 0.067 | 0.000 | 0.002 | 0.051 | 0.000 |
| Education level | Correlation coefficient | 0.054 | 0.052 | 0.062 * | 0.015 | −0.088 ** |
| | Sig. (2-tailed) | 0.081 | 0.093 | 0.045 | 0.624 | 0.005 |
| Month income | Correlation coefficient | 0.124 ** | 0.031 | 0.046 | 0.041 | 0.105 * |
| | Sig. (2-tailed) | 0.002 | 0.323 | 0.140 | 0.183 | 0.011 |
| Profession | Correlation coefficient | 0.104 ** | 0.049 | 0.060 | 0.095 * | 0.013 |
| | Sig. (2-tailed) | 0.001 | 0.115 | 0.053 | 0.035 | 0.671 |
| Distance | Correlation coefficient | 0.117 ** | 0.092 ** | 0.064 | 0.068 | −0.085 ** |
| | Sig. (2-tailed) | 0.001 | 0.003 | 0.055 | 0.059 | 0.006 |
| Living time | Correlation coefficient | 0.012 | 0.040 | 0.178 *** | 0.182 *** | 0.023 |
| | Sig. (2-tailed) | 0.515 | 0.202 | 0.000 | 0.000 | 0.462 |

Correlation is significant at three levels, * $p < 0.05$, ** $p < 0.01$, *** $p < 0.001$.

## 5. Discussion

Human societies can simultaneously threaten and be threatened by karst, which has been shaped by human–karst interactions and must also be mitigated from both directions [17]. Regulating karst via national legislation and administrative measures is one method currently used to manage this interaction and minimize its threats.

### 5.1. Policy-Oriented Conservation and Integration at Different Levels

Karst-related regulations and policies have been used in many countries since the mid-1980s [13,17,27,38]. In the United States, certain karst issues are often addressed via a state's administrative code [9] and local policy-based solutions, e.g., in Austin, Texas [39]. In China, policies have played important roles in land use/cover change since the 1970s in southwestern karst areas [10]. However, there are no unified or specialized karst heritage regulations or management law, as shown by the results above. Currently, karst conservation management is scattered among relevant laws, regulations, and policies in various fields at national and local levels. Policy-based solutions have been successful in some cases, e.g., to minimize the risks of water contamination. However, in many cases, karst heritage conservation is more of an afterthought or byproduct, usually in the course of setting rules for natural resource management or land planning. Some policies have been less effective in protecting karst heritage. In many cases, this failure is a result of vaguely defined goals, a lack of appropriate policy tools, weak or nonexistent enforcement, or poor conception or execution [39].

The present study indicated that the nature of problems involving human–karst interactions often requires solutions derived from more than one field. However, local regulatory bodies often have more narrowly focused areas of responsibility that problematize implementing an interdisciplinary approach. Many existing karst protection regulations have important flaws that hamper effectiveness due to a lack of integrating policies at different scales. There has been a tendency to consider political and cognitive factors as determinants of change in policy decisions; these factors are also relevant for handling conflicts. Different influences and inputs, e.g., the karst's physical characteristics,

regulation-writing experience, or the level of the actors' interest and input, can affect the regulation process. Some factors such as population, economics, and the nature of the local karst system or specific karst-related issues cannot completely explain the variety of the nature and structure of karst-related regulations between cities/provinces. With so many variables at work, uniformity is perhaps unattainable for karst heritage conservation regulations in different areas.

### 5.2. Governance Challenges and Stakeholder Participation

Karst heritage governance is one way to bring different sectors and actors, such as water conservancy, forestry, mining, agriculture, and urban infrastructure, together to share a conservation vision and develop a consensus for maintaining agreed-upon environmental, economic, and cultural values. A previous integrated conservation and development project study underlined the importance of strengthening institutions at national, provincial, and local levels; adopting a multi-layered and cross-sectoral governance approach; increasing stakeholders' coordination and capacity, especially through spatial planning activities; establishing long-term stakeholder partnerships; and effective action against illegal activities, including through law enforcement [40]. These are similar to the notion of a nexus that interlinks economic, environmental, and social systems [41]. This nexus approach is also a systematic process for both analysis and policy making to unpack the interdependencies between water, energy, food, and other linked systems, with the final aim of promoting cross-sectoral integration, sustainability, synergies, and resource use efficiency [36,42]. Enhancing connectivity in landscape governance requires coordination between multiple stakeholders across scale levels [33]. Because the wants and needs of stakeholders are varied, innovative modes of governance are required. These types of governance should be specifically tailored to address karst landscape management challenges [37]. To date, cross-sectoral challenges in Ethiopian forest and landscape restoration governance have been poorly addressed. For example, agricultural and environmental policy objectives, targets, and restoration mandates at the sub-national level are incoherent, and land use planning instruments make it difficult to negotiate trade-offs and find synergies between sectoral policy objectives [43].

In China, cross-sectoral coordination and multi-level network governance should be encouraged. The present network analysis showed the complexity of coordinating a large network for karst landscape governance (Figures 5 and 6). Although they have different objectives, governments at the national, provincial, and municipal levels have similar structures. This paper empirically presented the guiding importance of the national government's implementation of the karst landscape heritage governance network. We found that a broader and more strategic whole-of-government approach was adopted by the national government, while operational aspects (e.g., being effective actors) and partnerships were the foci of municipalities. In a large multi-level and multi-sectoral karst landscape heritage governance network, the national government is still dominant. This result is of significance for stakeholders' access to information and obtaining knowledge on coordination. Cooperation from the local to national levels and across sectors could make it possible to identify synergies, joint implementation pathways, and partnerships among different governments and other actors involved in karst heritage conservation. The potential of complementary and interconnected interventions would benefit all governments and stakeholders and increase the efficiency of resource use and capacities. For example, there are some successful reports in the fields of water governance, ecosystem corridors, resources, and biodiversity [37,44]. Our findings align with existing literature on the structure and role of local networks [45].

For karst heritage conservation, managers and practitioners have also highlighted governance effectiveness problems as a pervasive challenge. Although the effectiveness of multilevel governance and participatory processes on landscape conversation is not self-evident [11,33], our study shows the importance of governmental law and policy and the unsatisfactory participation and coordination of karst landscape heritage governance

in China. Complex karst heritage governance involving multiple stakeholders requires knowledge management, the involvement of local/regional authorities (as highly legitimated and connected actors within their local communities), a well-managed co-creation process to search for true collaboration, and specific rules and rule-making processes. The pre-existence of a participatory culture and multi-sectoral networks that are already engaged in landscape-related or environmental issues could also have a positive impact on the effectiveness of kart governance networks [46,47].

## 6. Conclusions

From the perspective of governance, the nexus governance approach presents a method to deal with the integration and interdependencies of the management of natural resources across sectors and actors [36,48]. Karst heritage sites are complex systems that requires multi-level, multi-sectoral, and multi-stakeholder nexus governance. Karst heritage landscape conservation plays two roles: one is to prevent and mitigate damage to local karst landscape systems and resources (e.g., water and groundwater supplies, tourist attractions), and the other is to maintain and/or improve ecosystem services affected by human activities in karst terrains [49]. To this end, karst heritage conservation should be accompanied by the strengthening of regulatory implementation, as well as promoting public awareness and partnerships. Karst-related regulations and ordinance implementation at multiple levels require greater integration and coordination. Although increasingly more national laws and local regulations have been enacted in China, these laws and regulations usually concern a single aspect of karst–human interactions, such as imposing strict control on water resources. In a complex system involving many actors at the national, provincial, and municipal levels, determining how to integrate their responsibilities effectively remains a challenge for karst heritage conversation. Therefore, multi-concern karst regulations that consider the impacts of new developments on karst landscapes, the integrity of different regulations, and various stakeholders are becoming increasingly necessary [50].

Considering existing karst heritage regulations and how stakeholders interact in China, we hope that this study provides the reader with a broad understanding of both the common issues in karst regulation and the regulatory toolbox that planners and regulators have at their disposal. This paper contributes to developing a link between governmental policy and karst heritage conservation effectiveness. In this work, we analyzed the multi-level policy related to karst conservation, identified different actors in large multi-sectoral governance networks of karst conservation stakeholders, and evaluated the perceived karst governance effectiveness by local residents. In the future, karst nexus governance should change from concept to action by focusing on the promotion of co-operative governance and procedures for coordinating environmental functions and ecosystems, including decentralization of the decision-making powers of governments. The successful factors related to karst governance should be systematically explored in a trans-disciplinary setting.

**Supplementary Materials:** The following supporting information can be downloaded at: https://www.mdpi.com/article/10.3390/heritage6060237/s1, Table S1: Key laws and regulations related to karst conservation and management in China; Table S2: Abbreviations of full list institutions.

**Author Contributions:** Conceptualization, methodology, data collection, formal analysis, writing, G.H.; investigation, X.Z. and G.H.; writing–review and editing, M.Y., L.Z. and L.S.; funding acquisition, G.H. All authors have read and agreed to the published version of the manuscript.

**Funding:** This work was supported by the National Natural Science Foundation of China (No. U21A2041); the National Key R & D Program of China (No. 2022YFC3802903); and Key Project of Guangxi Department of Science and Technology (Gui Ke AA20161004-04).

**Institutional Review Board Statement:** Not applicable.

**Informed Consent Statement:** All subjects gave their informed consent for inclusion before they participated in the study.

**Data Availability Statement:** Not applicable.

**Conflicts of Interest:** The authors declare no conflict of interest.

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
