# Peer review of "Karst Landscape Governance in the Guilin World Heritage Site, China"

_heritage, doi:10.3390/heritage6060237_

Round 1
Reviewer 1 Report
Review for article:
Karst Landscape Governance in Guilin World Heritage Site, China (authors: Guizhen He et al.)
1. Manuscript is maybe too long, as sometimes there are too long discussions.
2. Fig. 4 is somehow »heavy«, make sure you need it, as Fig 5 a-c is maybe enough.
3. Manuscript is probably usefull for other similar study sites and is a good base for future sustainable development of Guilin World Heritage Site.
4. Some additional remarques are in manuscript.

Author Response
Thanks for your detail comments.
1. Manuscript is maybe too long, as sometimes there are too long discussions.
Response: The discussion section is revised. It is a little longer in order to tell a whole story.
2. Fig. 4 is somehow »heavy«, make sure you need it, as Fig 5 a-c is maybe enough.
Response: Fig 4 and Fig 5 present the network relationship at different levels. They are both necessary.
3. Manuscript is probably usefull for other similar study sites and is a good base for future sustainable development of Guilin World Heritage Site.
Response: Thanks for your comments.
4. Some additional remarques are in manuscript.
Response: Thanks. We have revised accordingly. The language is edited.
Reviewer 2 Report
Very interesting and well-conducted contribution to karst management policy studies.
Apart from an error in the introduction to be corrected, only a few improvements and precisions are required or suggested (see details in the pdf file).
A regret all the same to have lost along the way the notion of nexus announced in the introduction, which disappeared from the discussion and the conclusion as explicit key-word.

Author Response
1.Very interesting and well-conducted contribution to karst management policy studies.
Response: Thanks for your comments.
2. Apart from an error in the introduction to be corrected, only a few improvements and precisions are required or suggested (see details in the pdf file).
Response: Thanks for your detail comments. We have revised the manuscript according to your comments.
3. A regret all the same to have lost along the way the notion of nexus announced in the introduction, which disappeared from the discussion and the conclusion as explicit key-word.
Response: In this version, we added some contents about the “nexus governance”. Please see discussion and conclusion section.
Reviewer 3 Report
This is a very interesting paper as it explains the very complex Chinese system of regulation, conservation, & management of the extensive Chinese karst. This is often difficult for karst workers outside China to understand. The choice of Guilin is excellent. The main issues are repetitive sentences & sections, especially the discussion, and Section 4.1. The Discussion needs to be clearer with respect to what the study shows rather than just repeating what is in previous sections. The conclusion misses some of the points made. Issues with English; see comments on annotated pdf of MS.

Author Response
1.This is a very interesting paper as it explains the very complex Chinese system of regulation, conservation, & management of the extensive Chinese karst. This is often difficult for karst workers outside China to understand. The choice of Guilin is excellent.
Response: Thanks for your comments.
2. The main issues are repetitive sentences & sections, especially the discussion and Section 4.1. The Discussion needs to be clearer with respect to what the study shows rather than just repeating what is in previous sections.
Response: We have revised the discussion section.
3. The conclusion misses some of the points made.
Response: We added some points in conclusion section.
4. Issues with English; see comments on annotated pdf of MS.
Response: We have polished the English via the MDPI editing service.
Round 2
Reviewer 3 Report
The upgrade of this paper has signifcantly improved it and it is now much more readable. It covers the complex area of cave and karst management in China. I have found one minor grammatical error (highlighted) than will need fixing. The supplementary information is fine.
